

# Methods of performance analysis in women's Australian football: a scoping review

Braedan R. van der Vegt[1], Adrian Gepp[1], Justin W.L. Keogh[2,3,4,5] and Jessica B. Farley[2]

[1] Centre for Data Analytics, Bond Business School, Bond University, Gold Coast, Queensland, Australia
[2] Faculty of Health Sciences & Medicine, Bond University, Gold Coast, Queensland, Australia
[3] Cluster for Health Improvement, Faculty of Science, Health, Education and Engineering, University of the Sunshine Coast, Sunshine Coast, Queensland, Australia
[4] Kasturba Medical College, Manipal Academy of Higher Education, Manipal, Karnataka, India
[5] Sports Performance Research Centre New Zealand, Auckland University of Technology, Auckland, New Zealand

Corresponding author
Braedan R. van der Vegt,
bvegt@bond.edu.au

## ABSTRACT

**Background**. The first women's Australian football (AF) professional competition was established in 2017, resulting in advancement in performance analysis capabilities within the sport. Given the specific constraints of women's AF, it is currently unclear what match-play performance analysis methods and techniques are implemented. Therefore, the aim of this scoping review was to describe and critically appraise physical, technical, and tactical performance analysis methods that have been employed in women's and girls' AF match-play.

**Methodology**. A systematic search was conducted on the 27th of June 2022 through five databases. Eligibility criteria were derived from the PCC framework with the population (P) of women and girls AF players, of any level of play; concepts (C) of interest which were measures, data, and methods related to the sport's physical, technical, and tactical performance; and the context (C) of methods that analysed any match-play performance. A narrative synthesis was conducted using extracted study characteristic data such as sample size, population, time period, collection standards, evaluation metrics for results, and application of thematic categorisations of previous sports performance reviews. Critical appraisal of eligible studies' methodologies was conducted to investigate research quality and identify methodological issues.

**Results**. From 183 studies screened, twelve eligible studies were included, which examined match-play through physical (9/12, 75%), technical (4/12, 33%), and tactical analysis (2/12, 17%). Running demands and game actions analysis were the most researched in senior women's AF. Research into junior girls' AF match-play performance has not been investigated. No research has been conducted on non-running physical demands, contact demands, acceleration, and tactical aspects of women's AF. All studies utilised either inferential statistics or basic predictive models. Critical appraisal deemed most studies as low risk of bias (11/12, 92%), with the remaining study having satisfactory risk.

**Conclusions**. Future research utilising increased longitudinal and greater contextual data is needed to combat the prominent issue of data representativeness to better characterise performance within women's and girls' AF. Additionally, research involving junior and sub-elite AF players across the talent pathways is important to conduct,

as it provides greater context and insight regarding development to support the evolving elite women's AF competition. Women's AF has been constrained by its resource environment. As such, suggestions are provided for better utilisation of existing data, as well as for the creation of new data for appropriate future research. Greater data generation enables the use of detailed machine learning predictions, neural networks, and network analysis to better represent the intertwined nature of match-play performance from technical, physical, and tactical data.

# INTRODUCTION

The Australian Football League Women's (AFLW) was established in 2017 as a national, elite competition. As a result, performance analysis capabilities within women's Australian football (AF) have increased due to the greater availability of data measuring match statistics, greater match coverage, access to facilities and sports science support, and player information (*Clarke et al., 2018*). This newfound capability for data collection enables the exploration of a wide array of performance areas of the women's game, including data analytics, which has become progressively more prevalent in performance analysis in sport (*Lord et al., 2020*).

Conducting performance analysis in sport allows for the characterisation and understanding of match-play systems, helping to better prepare athletes by quantifying the demands of a sport and informing coaching practices leading to increased performance (*Passos, Araújo & Volossovitch, 2017*). Previous performance analysis research in men's AF has explored physical performance and associated constraints (*Aughey, 2013*; *Gastin et al., 2013*; *Johnston et al., 2016*; *Johnston et al., 2012*), technical skill involvements (*Robertson, Back & Bartlett, 2016*; *Woods, 2016*), tactical considerations (*Alexander et al., 2019*; *Spencer et al., 2019*; *Taylor et al., 2020*; *Young et al., 2019a*) and hybrids of these areas (*Sheehan et al., 2021*; *Woods et al., 2016*; *Young et al., 2020*), using a variety of methods including machine learning such as through decision trees, bagging techniques, network analysis, and random forests (*Robertson, Back & Bartlett, 2016*; *Young et al., 2019b*; *Young et al., 2020*). Collectively, the findings of similar performance analysis research have the potential to improve women's AF performance characterisation by identifying key performance indicators and match-play patterns which may assist teams to better their player recruitment, coaching, player development, and gameday tactics (*Lord et al., 2020*).

Due to the relative infancy of the AFLW, it is currently unclear what performance analysis research has been conducted in women's AF. This presents challenges to women's AF practitioners and researchers regarding what evidence base they should use to guide industry best practices and what additional research needs to be prioritised as more varied and longitudinal data becomes available. This is relevant given that results produced by examination of men's match-play performance may be assumed to translate to women's

performance without further investigation in many sports despite noted differences in current match-play performance literature (*Emmonds, Heyward & Jones, 2019*). Such an assumption fails to account for the gender differences in human physiology and athlete profile, as well as access to sport science facilities and development pathways that differ between women's and men's sports (*Emmonds, Heyward & Jones, 2019*). Therefore, it is imperative that research is conducted with the specific objective of understanding women's performance to help alleviate this issue.

Past research of *Lord et al. (2020)* conducted a systematic review that investigated technical and tactical match-play performance analysis methods across multiple men's team-based invasion sports. The findings from this systematic review revealed themes of performance analysis methods that act as a framework to assist future research in these sports (*Lord et al., 2020*). The six main themes identified within this framework were game actions, dynamic game actions, movement patterns, collective team behaviours, social network analysis, and game styles (*Lord et al., 2020*). Applying this framework allows a holistic process-oriented approach to summarising performance analysis techniques, whether they have been applied appropriately, and provides a basis for appropriate future research recommendations by considering the individualistic nature of each team/player in a sport (*Lord et al., 2020*).

Physical performance is a prominent subject in other AF reviews (*Gray & Jenkins, 2010*; *Johnston et al., 2018*). These reviews have helped to understand physical match-play demands and results of previous physical performance analyses in men's AF. *Johnston et al. (2018)* looks at specific themes of physical research of men's AF, which provides a framework for current research in the area. Given its publication date, this review only briefly mentions the results of the beginnings of women's AF performance analysis pursuits, with no in-depth analysis undertaken nor investigation of methods of analysis employed (*Johnston et al., 2018*). This lack of specificity to women's AF in reviews, particularly when considering match-play performance and the differences that exist between women's and men's AF, indicates that a gap exists for a literature review of women's AF (*Clarke et al., 2018*; *Dwyer, Di Domenico & Young, 2022*).

With the increasing prominence of women's AF, as evident by the increase in participation of women of all levels in AF and the expansion of the AFLW competition (*AFL, 2020*; *Kleyn, 2021*), a review of women's AF performance analysis literature can guide a targeted, iterative approach which can be used to investigate performance within women's AF further and establish clear foundations for future research and practice to build upon. This is consistent with several other reviews on sports science research methodologies (*Hindle et al., 2020*; *Lord et al., 2020*; *Sarmento et al., 2014*). These reviews produced outcomes that assisted in identifying key under-researched themes, information and evidence bases for industry practitioners, areas of methodological concern, and potential future methodologies that can be applied. A review of the application of methods can allow for a greater consideration of the overall context surrounding women's AF, particularly considering the AFLW competition, which can be seen as a focal point for research going forward. Research methodologies applied in other sports or in the men's version of AF may not be currently available and/or entirely appropriate for exploration in women's AF.

The scenarios that the dynamic system of AF match-play present makes the selection of appropriate objective measures, methodologies, and modelling considerations within, unique compared to other sports (*Passos, Araújo & Volossovitch, 2017*). This includes selection of appropriate statistical modelling and data analytic approaches, such as descriptive statistics, inferential statistics, or predictive and advanced machine learning models, applied with AF-specific match-play performance outcomes as an objective.

As such, considering the context of what can be achieved with current resources at all levels-of-play in women's AF regarding data collection protocols and methodology selection to investigate match-play performance is critical to best allocate scarce resources. Understanding current methods of performance analysis could better characterise resource availability in practice and highlight gaps in the women's game research, which can begin to be alleviated in the future. Given the fast evolution of the women's iteration of the sport, it is expected that many physical, technical, and tactical relationships that have been described so far may not hold for an extended period of time, potentially being superseded shortly after any review, as more data becomes available. While review of results remains important to determining appropriate practice, it was deemed that more value would be derived from a review of methods applied currently. Consequently, a review of methods, their application, areas investigated, data utilised, and data analysis techniques may provide a longer lasting impact compared with a snapshot in time from a result-focused review.

Therefore, the primary aim of this scoping review was to describe methods of performance analysis that have been employed in women's AF match-play, taking a physical, technical, or tactical perspective. The secondary aim was to critically appraise the methodological quality of current literature. This review provides practical information for researchers, sports science practitioners, and data analysts working with women's AF players and guides future women's AF performance analysis research methodology and themes.

## MATERIALS AND METHODOLOGIES

### Registration

Consistent with the Preferred Reporting Items for Systematic Reviews and Meta-Analyses extension for Scoping Reviews (PRISMA-ScR) guidelines (*Tricco et al., 2018*), this systematic scoping review was registered with Open Science Framework on the 27th of March 2021 and amended on the 30th of January 2023. (https://osf.io/ufczd/ and https://osf.io/vxb8q)

### Data sources

A search strategy was created and adapted to the following five databases: MEDLINE, ProQuest Central, Scopus, SportDiscus, and Web of Science. The search was conducted from the earliest record of each database up to the 27th of June 2022. To ensure the thoroughness of the systematic search, grey literature was also searched for through the Proquest Dissertations database. The reference lists of eligible studies were also manually screened for additional relevant studies and reports.

**Table 1  Eligibility criteria used to screen studies.**

| Eligibility Criteria |
| --- |
| **Inclusion Criteria** |
| 1. Study examined a women's AF population of any age or level-of-play |
| 2. Study included either technical, tactical and/or physical performance measure |
| 3. Analysis of performance within the study is specific to match-play |
|     a. That is, analysis of player or team performance data within a match context through physical, technical, or tactical measures |
| **Exclusion Criteria** |
| 1. Study extracted not in English |
| 2. Studies where AF data/results indistinguishable from other sports data |
|     a. That is, grouped athlete results not coded by sport |
| 3. Studies where AF data/results indistinguishable between gender |
| 4. Cannot access full text |
| 5. Studies conducted through a literature or systematic review |

## Search strategy

The search strategy was created using the Population Concept Context (PCC) framework to define the research question under investigation (*JBI, 2015*). The population (P) of interest was women and girls AF players, of any age or level of play. The concepts (C) of interest were performance analysis measures and methods related to the sport's physical, technical, and tactical requirements. The context (C) included methods that explicitly analysed and reported any match-play performance characteristics. Due to the emergent nature of the research field, a broad set of search terms were selected relating to the PCC framework elements (see Supplementary Information 2).

## Inclusion/exclusion criteria

Studies were screened for eligibility by their title and abstract, and then full-text, using a set of inclusion and exclusion criteria created according to the PCC framework elements (see Table 1).

## Data management
### Selection process

All search returns were imported to EndNote (version X9; Thomson Reuters, Toronto, CA) for the removal of duplicates and the selection process. The pre-defined eligibility criteria were implemented independently by two reviewers in EndNote. All titles and abstracts were screened and categorised into bins of 'relevant', 'not relevant', and 'uncertain'. All 'uncertain' and 'relevant' studies from abstract screening were subjected to full-text reviews with the same set of criteria, applied by the same two reviewers independently. There were no disputes between the two reviewers over the inclusion of studies after the full-text screening. Both reviewers were not blinded to study authorship information throughout the screening process.

## Data extraction

One reviewer extracted individual study data, which a second reviewer reviewed to verify that no translation errors were made in the extraction process. The extraction of data was collated into a spreadsheet using Microsoft Excel. Data collected included metadata about the paper (authors, year, journal published), study type, population, contextual information of the data and methods used, study objective(s), and performance analysis methods, definitions, and outcome measures used.

## Critical appraisal

To assess methodological quality, two independent reviewers performed a critical appraisal of all included studies. The critical appraisal of the research methodology of included studies is essential to the review, despite it not typically being conducted in scoping reviews (*Munn et al., 2018*). It ensures that studies can be reliably used for reference while establishing a precedent for the appropriate application of methodology for practitioners. As the eligibility criteria for this scoping review included all study designs, the 16 questions presented by *Hindle et al. (2020)* were applied as it has been previously in other sport science systematic reviews (*Natera, Cardinale & Keogh, 2020*; *Nicol et al., 2022*; *West et al., 2022*). These questions were developed to address the possibility of different designs and assess both methodological quality and risk of bias (*Hindle et al., 2020*). Each question scored a 1 or 0 dependent on a yes or no answer, respectively, providing a total critical appraisal score of 0–16. Total scores were then calculated into a percentage and graded using the suggested thresholds of quality: low risk of bias (>66%), satisfactory risk of bias (34–66%), and high risk of bias (<33%) (*Hindle et al., 2020*). Disagreements on critical appraisal question scores were resolved by a third reviewer. The critical appraisal results for each eligible study were then considered when commenting on research questions, method, statistical appropriateness, sample selection, and final conclusions.

## Data synthesis

Studies were classified into themes by whether they addressed physical, technical, or tactical measures. Those deemed to address physical performance were further sub-categorised using a selection of physical performance themes previously identified by *Johnston et al. (2018)* that are specific to AF match-play. Categorisations were modified where themes were captured simultaneously in a performance analysis measurement. Technical and tactical studies were categorised into the performance analysis areas presented by *Lord et al. (2020)*. Explanations of the thematic categories utilised for the narrative synthesis are provided in Table 2.

Performance analysis methods utilised in included studies were deemed to use either descriptive statistics, inferential statistics, predictive modelling, or other modelling methods (including machine learning) to determine relationships of variables to match-play performance measures (*Lord et al., 2020*).

A narrative synthesis was conducted to summarise findings utilising the categorisations outlined in Table 2 in conjunction with individual study design, data characteristics, contextual variables, and critical appraisal results.

**Table 2  Explanation of performance categorisations.**

| Physical performance categories (*Johnston et al., 2018*) | |
|---|---|
| **Theme** | **Description** |
| Running demands | Measures of running performance often in categories of low-, moderate-, high-speed running, and sprinting as well as distance measures |
| Pacing and peak periods | Use of energy across matches and measuring periods of high intensity |
| Training load measures | Measures of physical performance and load in training tested relative to match-play performance |
| Player quality and experience | Measures of player performance by experience and perceived quality |
| Physical qualities | Anthropomorphic characteristics and association with performance tested prior to match |
|     Aerobic fitness | Tests of aerobic capabilities often through time trials and Yo-Yo tests |
|     Speed and acceleration | Tests of sprint and acceleration through individual and repeated short sprints |
|     Agility | In men's AF, the AFL have a standard change of direction test |
|     Strength and power | Measures of the muscular strength profile of athletes through tests such as 1 rep maximum squats, bench press, and counter movement jumps |
|     Body composition and anthropometry | Measures of height and body mass through both fat and lean mass |
| Match outcome | Physical performance relative between winning and losing team/players |
| Acceleration | Measures associated with acceleration |
| Match importance | Performance comparison with matches of increased importance (*e.g.*, finals matches) |
| Contact demands | Physical demands associated with game actions involving contact (*e.g.*, tackling) |
| **Technical/Tactical performance categories (*Lord et al., 2020*)** | |
| Game actions | Actions performed by teams and players and their association with success |
| Dynamic game actions | Examination of game actions with greater data capturing contextual factors informing each action |
| Movement patterns | Tactical movement patterns of players/ball in match-play |
| Social network analysis | Measures of team connectivity and cohesiveness through network analysis methodology |
| Collective team behaviours | Measures of team structure and movement |
| Game styles | Measure of team specific patterns that define match strategy |

# RESULTS

## Study selection

Figure 1 presents the stages of the search and screening processes. Following duplicate record removal, the five database and manual citation searches produced 183 records. The
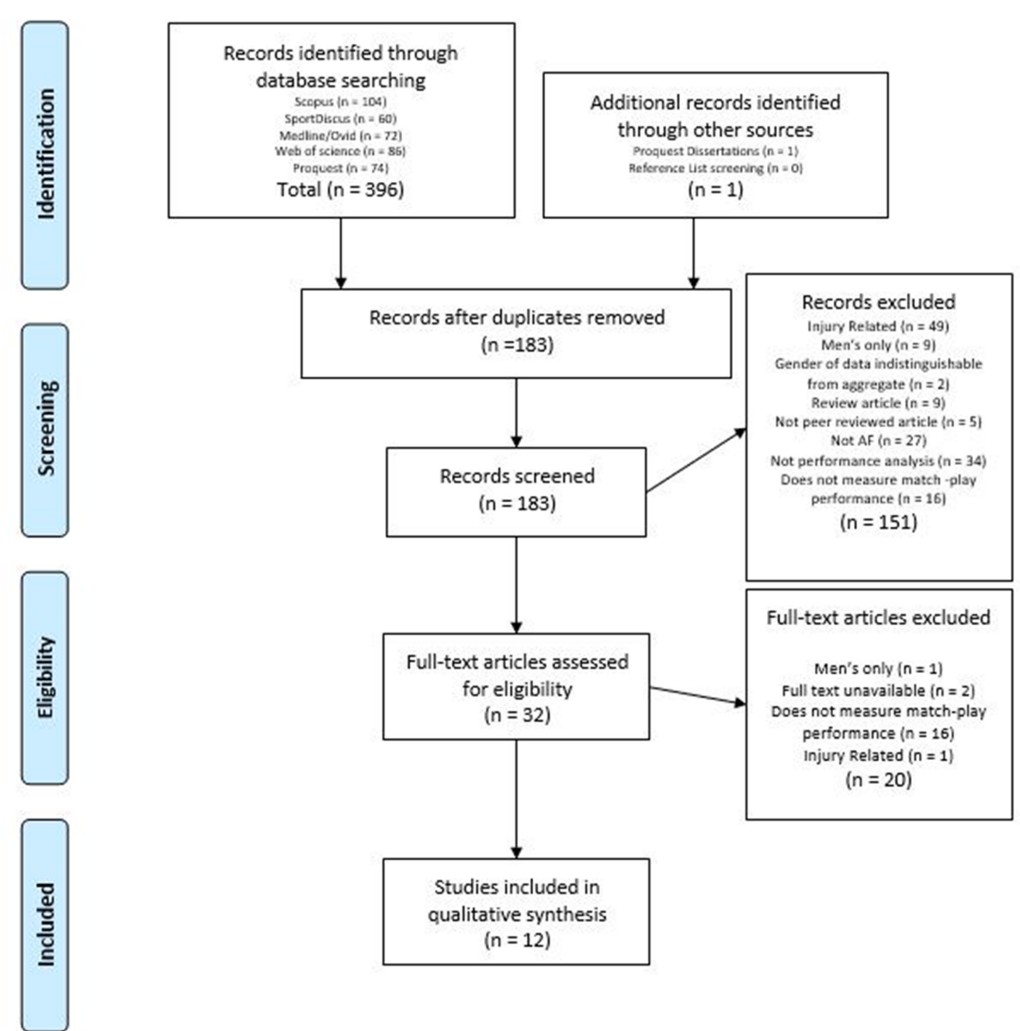

**Figure 1   PRISMA flowchart of the search process.** Abbreviation: AF, Australian Football.

screening process resulted in 12 studies included for analysis. The most prominent criteria for exclusion were studies with a lack of match-play measures and research objectives not analysing technical, tactical, or physical performance.

## Study characteristics

The 12 identified studies were classified to address themes of physical ($n = 9$, 75%), technical ($n = 5$, 42%) and tactical performance analysis ($n = 2$, 17%). Both tactical studies (*Black et al., 2019a*; *Cust et al., 2019*) also addressed technical performance, while two studies addressed both physical and technical measures (*Clarke et al., 2018*; *Clarke, Whitaker & Sullivan, 2021*). Investigation of the elite level of play was more prominent ($n = 7$) relative to the sub-elite level ($n = 4$), with one study presenting a comparison of these levels of play (*Clarke et al., 2019*). Only three studies (*Clarke, Whitaker & Sullivan,*

*2021*; *Cust et al., 2019*; *Dwyer, Di Domenico & Young, 2022*) used data from multiple seasons, with the remaining studies only using a single season sample ($n = 9$). All studies solely investigating a sub-elite level-of-play ($n = 4$) were conducted on 2016 season data, while elite level-of-play was analysed in the 2017 ($n = 6$), 2018 ($n = 3$), 2019 ($n = 2$), and 2020 ($n = 2$) seasons.

## Methods of match-play physical performance

Table 3 presents the studies that investigated physical performance measures in women's AF. All physical performance studies ($n = 9$) were of an observational nature. Traditional inferential statistics and effect sizes were employed to evaluate relationships between physical performance variables and match-play outcomes or comparison of different population groups in all studies ($n = 9$). For the purposes of this review, the theme sub-categories of distance, high-speed distance, and sprinting were condensed into one all-encompassing sub-category of running demands, given the high overlap of studies covering elements of each of these measures. Therefore, the physical performance analysis sub-categories utilised were running demands, pacing and peak periods of performance, training measures, player physical qualities (when analysed concerning match-play data in the form of aerobic fitness, speed and acceleration, agility, strength and power, or body composition and anthropometry), the effect of match outcome, player quality and experience, acceleration, match importance (*e.g.*, finals games), and demands associated with physical contact (*e.g.*, tackling) (*Johnston et al., 2018*). Included studies considered the running demands of the sport ($n = 9$), pacing and peak periods ($n = 5$), training measures that are related to on-field performance ($n = 2$), player quality and experience ($n = 2$), and player physical qualities measured outside match-play and tested for a relationship to match-play performance ($n = 3$) including tests of aerobic fitness ($n = 3$), tests of speed and acceleration ($n = 1$), and tests of strength and power ($n = 1$). Match outcome ($n = 1$) and match-play acceleration ($n = 1$) were also addressed. No study addressed match importance or contact demands. Subsets of physical qualities of agility and anthropometry were also unaddressed. Two differing thresholds of running speed were utilised across the nine studies examining physical performance (Table 3).

## Methods of match-play technical and tactical performance

Table 4 presents included studies that measured technical and tactical performance analysis qualities in women's AF. The two studies that conducted tactical analysis used a predictive model, while the other purely technical studies ($n = 3$) used inferential statistics. The primary categorisation of the analyses was game actions ($n = 5$), with one study also investigating technical and tactical elements of game styles. Dynamic game actions, collective team behaviours, social network analysis, and movement patterns were not addressed in any of the included studies.

## Critical appraisal of eligible studies

The critical appraisal scores are shown in Table 5. Eleven studies were deemed to have a low risk of bias, while one study was deemed to be of a satisfactory risk (*Thornton et al., 2020b*). Cohen's Kappa analysis found a fair level of agreement between the two reviewers
**Table 3  A summary of the characteristics of studies examining physical qualities of women's Australian Football players.**

| Author (Year) | Theme Sub-Category | Study Objective | Level-of-Play | Position(s) considered? | Seasons | Data Sample | Evaluation Metric | Running Speed Standards used |
|---|---|---|---|---|---|---|---|---|
| *Black et al. (2018a)* | Running Demands, Physical Qualities –Aerobic Fitness + Strength and Power + Speed and Acceleration, Player Quality | To highlight the physical qualities that discriminate selected and nonselected female AF players, investigate activity profiles of female AF players, and gain an understanding of the influence of physical qualities on performance in female AF | Sub-elite QAFLW (Academy Selected vs Non-academy selected) | Yes | 2016 | Academy Selected: 22 players Non-academy selected: 27 players | Inferential Statistics (Effect size) | LSR (0-10 km h$^{-1}$) MSR (10–14.94 km h$^{-1}$) HSR (>14.94 km h$^{-1}$) |
| *Black et al. (2018b)* | Running Demands, Physical Qualities –Aerobic Fitness, Pacing, Training | To investigate the influence of physical fitness on peak periods of match-play. | Sub-elite (QAFLW) | Yes | 2016 | 43 players across 3 teams (180 match files) + Pre-season Yo-Yo IR1 results | Inferential Statistics (Effect size) | Same as *Black et al. (2018a)* |
| *Black et al. (2018c)* | Running Demands, Physical Qualities –Aerobic Fitness, Pacing | (1) to compare activity profiles of on-field bouts between rotated and whole-quarter player performances, (2) to identify the changes in running performance during different on-field bout durations, and (3) to investigate the influence of YoYo IR1 performance on activity profiles. | Sub-elite (QAFLW) | No (Midfielders only) | 2016 | 22 players across 3 teams (97 match files) | Inferential Statistics (Effect size) | Same as *Black et al. (2018a)* |
| *Black et al. (2019b)* | Running Demands, Match outcome, Pacing | To identify differences in pacing strategies and activity profiles among female AF match-play, based on game outcome and opponent rank. | Sub-elite (QAFLW) | Yes | 2016 | 35 players across 3 teams (14 matches) | Inferential Statistics (Effect size) | Same as *Black et al. (2018a)* |
| *Clarke et al. (2018)* | Running Demands, Pacing | To describe the physical and technical demands of AFLW match-play for different positional groups. A secondary aim was to examine the time course changes in running performance during AFLW match-play. | Elite (AFLW) | Yes | 2017 | 26 players from 1 club (143 match files) | Inferential Statistics (Effect size) | HSR (>14.4 km h$^{-1}$) VHSR (>18.0 km h$^{-1}$) Sprinting (>20.0 km h$^{-1}$) |
| *Clarke et al. (2019)* | Running Demands, Player Quality | To quantify and compare the match running demands of the AFLW and VFLW competitions, with reference to playing positions and playing quarters | Elite (AFLW) vs Sub-elite (VFLW) | Yes | 2017 | AFLW: 27 players from 1 club (91 match files) VFLW: 36 players from 1 club (263 match files) | Inferential Statistics (Effect size) | LSR (0–14.4 km h$^{-1}$) HSR (same as *Clarke et al., 2018*) VHSR (same as *Clarke et al., 2018*) Sprinting (same as *Clarke et al., 2018*) |
| *Clarke, Whitaker & Sullivan (2021)* | Running Demands | To observe the position-specific peak movement demands of AFLW players and assess whether any seasonal changes have occurred in movement- or performance-based metrics over the initial three years of competition | Elite (AFLW) | Yes | 2017–2019 | 44 players from 1 club (21 matches over 3 seasons) | Inferential Statistics (Effect size) | HSR (same as *Clarke et al., 2018*) |
| *Thornton et al. (2020a)* | Running Demands | To provide an overview of the externally measured movement characteristics of AFLW competition, and the within-subject variability between matches. | Elite (AFLW) | Yes | 2020 | 28 players from 1 club (7 matches, 140 match files) | Inferential Statistics (Effect size) | HSR (same as *Clarke et al., 2018*) VHSR (same as *Clarke et al., 2018*) |
| *Thornton et al. (2020b)* | Running Demands, Pacing, Acceleration, Training | To (a) provide an overview of the weekly externally measured training loads across the AFLW season, which will assist in the preparation of athletes for competition and…(b) …analysis of the training undertaken, where the distribution of volume accumulated within training drills relative to the 1min MM intensity of matches was established. | Elite (AFLW) | No | 2020 | 28 players from 1 club (161 match files) | Inferential Statistics (Effect size) | HSR (same as *Clarke et al., 2018*) VHSR (same as *Clarke et al., 2018*) |

**Notes.**

AF, Australian football; AFLW, Australian Football League Women's; Yo-Yo IR1, Yo-Yo Intermittent Running Level 1 test; LSR, Low-Speed Running; Moderate-Speed Running; HSR, High-Speed Running; VFLW, Victorian Football League Women's; VHSR, Very High-Speed Running.

All studies used S5 unit sampling at 10Hz from Catapult Sports for data collection with validity and reliability previously verified in *Johnston et al. (2014)*.

van der Vegt et al. (2023), *PeerJ*, DOI 10.7717/peerj.14946

**Table 4** A summary of the characteristics of studies examining technical and tactical qualities of women's Australian Football players and teams.

| Author (Year) | Theme sub-category | Study objective | Number of variables investigated | level-of-play | position(s) considered? | Seasons | Data sample | Model type | Evaluation metric |
|---|---|---|---|---|---|---|---|---|---|
| | | | **Tactical/technical studies** | | | | | | |
| Black et al. (2019a) | Game Actions | To investigate the relationship between technical involvements and (1) winning margins, (2) losing margins and (3) ladder position in the national AFLW competition. | 13 basic team match-play variables (K, H, D, DE%, CP, UP, M, CM, CG, T, I50, GA%, I50:G) | Elite (AFLW) | No | 2017 | 26 matches in win/loss subsets (2 draws removed) | Predictive (CHAID decision Trees + Logistic Regression) | Significance tests of tree nodes |
| Cust et al. (2019) | Game Actions + Game Styles | To evaluate the relationship of AFLW athlete skill performance indicator distributions, to explain match quarter outcomes during the 2017 and 2018 seasons. Secondly, this study aimed to compare quarter outcome model error rates from separate machine learning approaches, based on the varied input feature set variables. | Multiple datasets with 13 basic/relative between teams variables + Feature distribution generating additional 11 new variables reflecting individual player contribution per base variable (H, CP, UP, UM, CM, CG, T, I50, HO, LK, SK, IK, K:HB) | Elite (AFLW) | No | 2017-2018 | 154 players from 7 clubs across 56 matches | Predictive (Generalized Estimating Equations + Decision Trees) | Model selection on Mean Absolute Error (MAE) on test data |
| | | | **Technical Studies** | | | | | | |
| Clarke et al. (2018) | Game Actions | To describe the physical and technical demands of AFLW match-play for different positional groups. A secondary aim was to examine the time course changes in running performance during AFLW match-play. | Nine basic player match-play variables by position (K, H, CP, UP, M, I50, R50, G, T) | Elite (AFLW) | Yes | 2017 | 26 players from 1 club (143 match files) | Effect Sizes only | Inferential Statistics (Effect size) |
| Clarke, Whitaker & Sullivan (2021) | Game Actions | To observe the position-specific peak movement demands of AFLW players and assess whether any seasonal changes have occurred in movement- or performance-based metrics over the initial three years of competition | Same as Clarke et al. (2018) (K, H, CP, UP, M, I50, R50, G, T) | Elite (AFLW) | Yes | 2017–2019 | 44 players from 1 club (21 matches over 3 seasons) | Linear Mixed Effects Model(s) | Inferential Statistics (Effect size) |
| Dwyer, Di Domenico & Young (2022) | Game Actions | To (1) establish normative values for a relatively wide range of team technical PIs in the AFLW, (2) compare team technical performance between men's and women's leagues, (3) identify apparent trends of change in technical performance over the first three seasons of AFLW and (4) assess the associations between technical performance and match outcome. | 21 basic team match-play variables (K, H, D, DE%, CP, UP, M, CM, M50, CG, HO, CL, CC, SC, T, I50, R50, FF, FA, 1%, B) | Elite (AFLW) | No | 2017–2019 | 96 matches from 10 clubs | Significance testing + Pearson Correlations | Significance of difference to AFL data and between seasons + Significance of correlation of statistics to match margin |

**Notes.**

All datasets are derived from the Champion Data collection service that has previously been deemed to be reliable (*Robertson, Back & Bartlett, 2016*).

AFLW, Australian Football League Women's; K, Kicks; H, Handballs; D, Disposals; DE%, Disposal Efficiency; CP, Contested Possession; UP, Uncontested Possession; M, Mark; CM, Contested Mark; CG, Clangers; T, Tackles; I50, Inside 50; GA%, Goal Accuracy %; I50:G, Inside 50:Goal ratio; UM, Uncontested Mark; LK, Long Kick; SK, Short Kick; K:HB, Kick: Handball Ratio; R50, Rebound 50; G, Goals; M50, Marks Inside 50; HO, Hitouts; CL, Clearances; CC, Centre Clearances; SC, Stoppage Clearances; FF, Frees For; FA, Frees Against; 1%, One percenters; B, Bounces.

($\kappa = 0.334$). Full agreement was achieved between reviewers on critical appraisal scores after further consultation. The most notable methodological issue was regarding the representativeness of the data sample to the population (Question 2.3), with a consistent theme of data samples not being representative of the overall population expressed in the aim of the studies. Multiple studies also failed to meet the criteria of how the study size was arrived at (Question 2.4) and the explicit criterion for inclusion of participants (Question 2.1).

## DISCUSSION

The purpose of this scoping review was to describe methods of performance analysis that have been employed in women's AF, taking a physical, technical, or tactical perspective. The secondary aim was to critically appraise the methodological quality of current literature. Gaps can be identified in whole themes of research within physical performance, and particularly in technical and tactical performance. Even the most established themes in these areas have room for further exploration when considering level-of-play. The critical appraisal highlighted an issue surrounding data representativeness of studies. Such a focus on research methodologies has important implications for both sports science researchers and practitioners by highlighting research gaps and issues surrounding access to detailed data and statistical expertise.

Exploration of data samples shows that only the elite and sub-elite level-of-play populations have been analysed, leaving the junior level-of-play unexplored. Performance measures in junior levels of the sport covering all three areas of performance analysis should be analysed as it provides benefits to junior levels by assisting in developing a better understanding of performance while creating better preparation practices for players drafted to the elite level, as it has in men's AF (*Robertson, Woods & Gastin, 2015*; *Woods et al., 2016*). It is recommended that the AFLW and state governing bodies prioritise standardised data collection to allow comparison between levels of play and enable replication studies in the women's game.

### Methods of match-play physical performance

The most prominent area of inquiry was physical performance, with all eligible studies in this area investigating running demands, as represented by GPS data. Physical performance research, despite its relative prominence, is not exhaustive, with talent identification, acceleration, training, match outcome, player quality, and physical qualities of strength, speed, and acceleration only assessed in one or two studies. Additionally, contact demands, match importance, agility, and anthropometry were not investigated in the current literature. Notably, in identified studies there is a lack of longitudinal data in all physical performance areas. As such, understanding physical performance analysis across key themes in women's AF is still in its infancy with many themes unexplored. Explored themes also have greater capacity for re-investigation using current research designs with the addition of more robust samples. It is noteworthy that all physical qualities investigations that involved measures from field-based testing as a factor

**Table 5  Application of critical appraisal tool on eligible studies.**

| Author (Year) | 1.1 | 1.2 | 1.3 | 2.1 | 2.2 | 2.3 | 2.4 | 3.1 | 3.2 | 3.3 | 3.4 | 4.1 | 4.2 | 4.3 | 4.4 | 4.5 | Score | % | ROB |
|---|---|---|---|---|---|---|---|---|---|---|---|---|---|---|---|---|---|---|---|
| Black et al. (2018a) | 1 | 1 | 1 | 0 | 1 | 0 | 0 | 1 | 1 | 1 | 1 | 1 | 1 | 1 | 1 | 1 | 13 | 81% | L |
| Black et al. (2018b) | 1 | 1 | 0 | 0 | 1 | 0 | 0 | 1 | 1 | 1 | 1 | 1 | 1 | 1 | 1 | 1 | 12 | 75% | L |
| Black et al. (2018c) | 1 | 1 | 1 | 1 | 1 | 0 | 0 | 1 | 1 | 1 | 1 | 1 | 1 | 1 | 0 | 1 | 13 | 81% | L |
| Black et al. (2019a) | 1 | 1 | 1 | 1 | 1 | 1 | 1 | 1 | 1 | 1 | 0 | 1 | 1 | 1 | 1 | 1 | 15 | 94% | L |
| Black et al. (2019b) | 1 | 1 | 1 | 1 | 1 | 0 | 1 | 1 | 1 | 1 | 1 | 1 | 1 | 1 | 1 | 1 | 15 | 94% | L |
| Clarke et al. (2018) | 1 | 1 | 1 | 0 | 1 | 0 | 0 | 1 | 1 | 1 | 1 | 1 | 1 | 0 | 1 | 1 | 12 | 75% | L |
| Clarke et al. (2019) | 1 | 1 | 1 | 1 | 1 | 0 | 1 | 1 | 1 | 1 | 1 | 1 | 1 | 1 | 1 | 1 | 15 | 94% | L |
| Clarke, Whitaker & Sullivan (2021) | 1 | 1 | 1 | 1 | 0 | 0 | 1 | 1 | 1 | 1 | 1 | 1 | 1 | 1 | 1 | 1 | 14 | 88% | L |
| Cust et al. (2019) | 1 | 1 | 1 | 1 | 1 | 1 | 1 | 1 | 1 | 1 | 0 | 1 | 1 | 0 | 1 | 0 | 13 | 81% | L |
| Dwyer, Di Domenico & Young (2022) | 1 | 1 | 1 | 1 | 1 | 1 | 1 | 1 | 1 | 1 | 1 | 1 | 1 | 1 | 1 | 1 | 15 | 100% | L |
| Thornton et al. (2020a) | 1 | 1 | 1 | 1 | 1 | 0 | 1 | 1 | 1 | 1 | 1 | 1 | 1 | 1 | 1 | 0 | 14 | 88% | L |
| Thornton et al. (2020b) | 1 | 1 | 1 | 0 | 1 | 0 | 0 | 1 | 1 | 1 | 1 | 1 | 1 | 0 | 1 | 0 | 11 | 69% | S |
| Total | 12 | 12 | 11 | 8 | 11 | 3 | 7 | 12 | 12 | 12 | 10 | 12 | 12 | 9 | 11 | 9 | | | |
| | 100% | 100% | 92% | 67% | 92% | 25% | 58% | 100% | 100% | 100% | 83% | 100% | 100% | 75% | 92% | 75% | | | |

**Notes.**

Critical Appraisal Questions: (1.1) study design is clearly stated; (1.2) the objectives/purpose of the study is clearly defined; (1.3) the design of the study adequately tests the hypothesis; (2.1) the criteria for the inclusion of participants is clearly described; (2.2) the characteristics of the population is clearly described; (2.3) the study sample is representative of the population intended to the study; (2.4) a description of how the study size was arrived at is provided; (3.1) the testing methods are clearly described; (3.2) the measurement tools used are valid and reliable; (3.3) the statistical methods used well described; (3.4) the statistical tests used to analyse the data are appropriate; (4.1) the results are well described; (4.2) the information provided in the paper is sufficient to allow a reader to make an unbiased assessment of the findings of the study; (4.3) confounding factors are identified; (4.4) sponsorships/conflicts of interest are acknowledged; (4.5) any limitations to the study are identified. Note: the risk of bias score for an article (given as a percentage) is calculated through the addition of the score from each criterion being met divided by the maximum possible score across all criteria (16), multiplied by 100. L low risk of bias (67–100%), S satisfactory risk of bias (34–66%), H high risk of bias (0–33%).

potentially influencing match-play performance, have all been conducted only on a sub-elite level. This represents a major literature gap with scope for tests of aerobic fitness (*e.g.*: Yo-Yo tests), speed and acceleration (*e.g.*: short sprint tests), agility (*e.g.*: AFL standard agility test), strength (*e.g.*: relevant repetition maximum lifts), and anthropometry measures (*Johnston et al., 2018*) all able to be evaluated for a relationship to match-play performance in the elite AFLW competition. Identification of such relationships would provide important data for sport scientists and strength and conditioning coaches, informing what constitutes more appropriate physical fitness assessments and overall training prescriptions.

Data synthesis indicated a discrepancy exists in the definition of high-, very high-speed running (HSR & VHSR) and sprinting measures across the studies. Defining the measures at different thresholds for HSR, VHSR, and sprinting may confuse the interpretation of results and alter the statistical significance of running demands to match-play performance. While two studies (*Black et al., 2018a*; *Clarke et al., 2018*) cited the work of *Bradley & Vescovi (2015)* as the basis for their definition, the initial study's definition was presented as a range rather than a specific threshold, leading to a discrepancy between the two interpretations. In future research, an exact set threshold and clear definition specific to women's AF is needed to ensure generalisability and enable a better comparison of results across studies.

Physical running performance data collection for all studies utilised the Catapult Sports Optimeye S5 unit (Catapult Sports, Victoria, Australia) sampling at 10 Hz. This has previously been deemed to be reliable for this purpose and superior to collection through 1 and 5 Hz units (*Johnston et al., 2014*). Some studies also chose to report other measures of reliability including mean horizontal dilution of precision (HDOP) and mean satellite availability, as well as reporting common practices like that of using the same unit per player across seasons to reduce inter unit error (*Clarke et al., 2018*; *Clarke, Whitaker & Sullivan, 2021*; *Thornton et al., 2020a*). Reporting of these metrics and following these collection protocols should continue across all future studies with only four papers not reporting these metrics (*Black et al., 2018a*; *Black et al., 2018b*; *Black et al., 2018c*; *Black et al., 2019b*), although all acknowledged the importance of collecting reliable and valid GPS data.

All studies investigating physical performance used traditional statistical tests of significance. The prevalence of this testing method can be somewhat limiting in both conclusions able to be drawn and on the singular perspective of statistical significance that it provides, depending on the data available (*Lord et al., 2020*). It was also observed that all studies of physical performance utilised effect sizes within their statistical analysis. While effect sizes as a means of interpreting magnitude allows for a more nuanced under-standing, issue can be taken on testing conducted without sufficient data or background context that is required in a sport where player and team action is highly conditional on surrounding match context. Using traditional statistical tests in isolation does not fully consider the multi-factorial aspects of match-play, leaving inter-relationships between physical performance metrics uncaptured. Linear cause–effect relationships, while having

a place for investigation in the literature as a basis, need to be expanded upon to begin to fully grasp the intricacies involved in match-play performance.

Performance analysis literature has shifted to present match-play in team-invasion sports as a set of 'dynamic systems' represented by the constraints or context surrounding each action (*Travassos et al., 2013*). While confounding factors were addressed through positional consideration in most included studies examining physical performance, the extent to which confounding factors have been captured in data is an issue. The incorporation of further confounding factors, such as weather, opposition strength, phase of play, timing of action, interaction with opposition, and position on the field, is needed to better represent the true running demands of match-play (*Travassos et al., 2013*). For example, significant differences were found between match phases of play (*e.g.*: team with or without the ball, stoppage of play) across high- and low-speed running, and acceleration in the AFL (*Rennie et al., 2020*), suggesting that the phase of play is a key contextual consideration that may be applicable to the analysis of physical performance measures within women's AF. The further collection and incorporation of data capturing the effects of these contextual factors presents a logical next step for further exploration of existing themes and studies for women's AF. Doing so would provide justification as to why a player has produced a physical output, as well as mitigate confounding relationships that may lead to inappropriate performance practices being implemented.

Overall, future research from a physical perspective should investigate physical performance over a longer period, mapping the potential development of AFLW athletes as more time is spent under an elite sport program. No testing of the link between physical qualities of players (*e.g.*, aerobic fitness, speed and acceleration, agility, strength and power, and anthropometry) and match-play performance has been conducted at the elite level representing a key area that could inform improved training design and preparation practices.

## Methods of match-play technical performance

Of the five studies to address technical skills, reporting effect sizes between playing positions were the method of choice for two studies (*Clarke et al., 2018*; *Clarke, Whitaker & Sullivan, 2021*) and reporting of *p*-value significance the choice of another (*Dwyer, Di Domenico & Young, 2022*). In contrast, the other two studies used predictive model methods relating match statistics of players and teams to success within games and seasons (*Black et al., 2019a*; *Cust et al., 2019*). As an exploratory examination of the technical factors deterministic of match success, the predictive studies are reasonable to use as an initial indication of important game actions within women's AF. Despite this, there is a lack of appropriate provisions within modelling practices to make declarations of relationships with any certainty or be reflective of anything more than a snapshot in time in a fast-developing environment. This is largely symptomatic of the scarcity in length and contextual information of technical actions within available data applied currently. In the modelling of *Black et al. (2019a)*, results were not verified on a holdout test set or cross-validated, a standard practice in modelling research (*James et al., 2013*). Potential flaws in the variable selection of *Black et al. (2019a)*, as highlighted by *Cust et*

*al. (2019)*, led to models built with an inherent bias using data that can be seen as a direct determinant of match success. Repeating similar methodology with the removal of these highly predictive variables over multiple seasons of match-play would enable greater robustness and validity of future findings.

The application of generalised estimating equations within *Cust et al. (2019)* was also not verified on test data, unlike the decision trees in the same analysis. Consequently, caution should be used when drawing conclusions about technical relationships to inform current match-play performance. Match-play technical skills may be considered underdeveloped within the first seasons of the competition. These were the seasons of focus in all previous studies that model technical data within match-play, a further important consideration when interpreting past results for application on the current AFLW competition. Future research re-evaluating previous technical relationships should be sought, aiming to capture and understand the development of the competition since inception. The methodology used should also be carefully considered to enable the capture of this development as has been performed in men's AF literature (*Woods, Robertson & Collier, 2017*).

The beginnings of this research into the evolution of the AFLW can be seen in *Dwyer, Di Domenico & Young (2022)* who investigated how some key match statistics may have changed over the initial three years (2017–19) of the competition. In addition, *Dwyer, Di Domenico & Young (2022)* also made a comparison to the men's game from a technical standpoint scaled by the time of play for each competition. While this represents a means of fairer comparison, total match times still do not account for differences in 'time on' rules leading to less time in play in the AFLW that will effect comparison in this manner (*Australian Football League Women's, 2021*). Twenty-seven variables were analysed for their correlation to match success in the form of match margin, six of which are functions of scoring, meaning that twenty-one unbiased statistics were tested. The significance testing of the correlations of the twenty-one statistics to match margin used is a univariate investigation of variable importance, meaning that results should be taken with caution as many confounding factors could potentially affect this significance.

The influence of confounding factors, such as opposition played, variation of key statistics over time, and team strategy should be considered in these results as well as in any future longitudinal study design to produce a better representation of the key variables affecting match-play performance. More intensive methodologies, including machine learning, can be implemented in the future when appropriate to assist in the incorporation of this additional data. While captured in aggregate form in these analyses (*Black et al., 2019a*; *Cust et al., 2019*), each team's game style may alter which game actions are prioritised to achieve match success. As a result, future research helping to quantify game styles can allow improved interpretation of important technical actions representative of these game styles.

## Methods of match-play tactical performance

Given the dearth of tactical performance analysis in the women's AF literature, there is a large amount of room for exploration within this area. While all of the six tactical

domains of the *Lord et al. (2020)* framework have been investigated in men's AF, only investigations of game actions (*Black et al., 2019a*; *Clarke et al., 2018*; *Clarke, Whitaker & Sullivan, 2021*; *Dwyer, Di Domenico & Young, 2022*) and game styles (*Cust et al., 2019*) have been conducted in women's AF. All six themes being applied in men's AF suggests that the 'dynamic system' of AF gameplay allows for methodologies in each of these themes to be investigated in women's AF. Whether they are capable of application in the current women's AF environment is another question, due to current differences in access to high granularity data available between men and women's AF. As indicated by observations surrounding the primary game actions studies (*Black et al., 2019a*; *Cust et al., 2019*), the theme has scope for greater exploration by considering the variability introduced by the competition's development and contextual factors on performance.

*Cust et al. (2019)* can be seen as the beginnings of the tactical investigation of the women's game, as the study sought to address the strategies employed by a team that results in success. The conclusion that the performance of the highest statistic accumulating players in a range of measures is paramount to team success is a tactical insight, although it does need further context and studies to verify this claim fully. A precedent that has been set for greater investigation of this tactical output within the men's sphere, is through network analysis utilising more granular data, allowing the mapping of player passing within a team *via* who is passing and receiving the ball (*Young et al., 2020*).

Other themes in the tactical categorisations of men's AF including movement patterns (*Sheehan et al., 2021*), collective team behaviour (*Alexander et al., 2019*), social network analysis (*Young et al., 2020*), and holistic mapping of game styles (*Woods, Robertson & Collier, 2017*), have not been examined within women's AF. The first three of these themes require the notation or collection of additional data beyond what is currently available to researchers of women's AF. Movement patterns and social network analysis would require the availability of spatio-temporal, notated match statistics to capture tactical match-play information; data that are currently captured and used in the AFL (*Spencer et al., 2019*; *Young et al., 2019a*) but unused in women's AF literature. This is possibly due to such data being unavailable in the AFLW or, if available, being beyond the means of many clubs to collect or purchase. Collective team behaviour requires GPS coordinates of all players on a team or whole game to be represented and has been conducted on a proof-of-concept basis in men's AF (*Alexander et al., 2019*; *Spencer et al., 2019*). This may be beyond the reach of current facility constraints of the AFLW, given the requirement of non-aggregated, dynamic game action data being available, particularly with more foundational objectives still unexplored (*Alexander et al., 2019*). Therefore, with current data provisions in mind, further research of game actions to reflect the development of match-play with additional context and an investigation of game styles should be prioritised. Strides should be made by both clubs, by allocating more resources to self-collection, and the league, by replicating AFL data collection in the AFLW, equalising the availability of data.

Due to the nature of tactical analysis, a prohibitive limitation of studies is that the most common form of data available, aggregate statistics, do not possess the granularity to give greater insights into match-play. It is, therefore, understandable that studies

have been limited in scope by the data that is currently available due to the recency of the AFLW competition. Further exacerbating these limitations are the relatively short seasons, and scarce analysis of multiple years of data in current studies. Additionally, the onus of research would be to produce more exploratory studies to give quicker, interpretable insights given the newness of competition, meaning that further data generation processes required to be implemented by teams or the competition have not been prioritised. Future research should seek to implement a data collection protocol that captures game actions with greater granularity, as well as greater contextual information surrounding each action. Specifically, future research is needed to determine what contextual information is important to capture.

The availability and granularity of data remain the greatest limitations within both technical and tactical analysis, which ultimately dictates whether intensive methodologies employed in men's AF and other sports can be employed for future research in women's AF. While most applicable to tactical analysis, the greater inclusion of contextual factors that determine match-play phenomena is a gap in the majority of the literature, as player positional consideration has been the only external factor considered in most cases (*McGarry, 2009*). In a holistic sense, the constraints that each sub-category (physical, technical, tactical performance) imposes upon the other can be incorporated to create more comprehensive models that better reflect match-play circumstances and the influence the three areas have on each other. Besides the natural overlap of technical and tactical performance modelling, no research has yet combined all elements to give a better representation of the multifaceted nature of player and team performance. Precedents exist that enable this information to be incorporated, such as neural networks used by *Watson et al. (2020)* that can allow for the data from physical, technical, and tactical performance to be utilised in one model allowing intertwined relationships of match-play performance demands to be reflected. Use of these models are still distant, given the intensive data required to perform this modelling in a robust manner. Future research could expand to begin to follow the precedent of the ecological dynamics approach of representing the complex system of match-play performance, while other constraints on match-play performance, such as what a player perceives in a moment in time, can also begin to be investigated. This approach can give the best representation of the requirements for performance and produce more representative training designs for better preparation (*Passos, Araújo & Volossovitch, 2017*; *Vilar et al., 2012*).

Further data generation to enable future research should also be considered by the league and teams. However, given current limitations in facilities in the competition, it may not be an immediate focus with current data able to be further utilised. Greater resources allocated by teams toward data collection as well as an equalisation of data availability in the AFLW relative to the AFL either through the primary collector, Champion Data or the league itself, should be sought. All technical and tactical data utilised in current research has been from the Champion Data collection service, which has previously been verified in its reliability and accuracy (*Robertson, Back & Bartlett, 2016*).

On the objective of greater use of current data, there is still room for more studies (that can still use traditional statistical hypothesis testing) as relationships established from these analyses can dictate the objectives of further research. An example of this is that more Inside 50s has been shown to lead to team success in current literature (*Black et al., 2019a*). In this case, a more process-oriented approach (*Passos, Araújo & Volossovitch, 2017*; *Vilar et al., 2012*) can be sought through analysis of how the Inside 50 statistic is attained in match-play, rather than overall match success being used as the performance outcome objective which can miss key details and trends.

Technical and tactical match-play performance remains only measured in relatively simplistic research designs, with all themes of investigation (game actions, dynamic game actions, movement patterns, social network analysis, collective team behaviours, game styles) able to be either researched for the first time or done with a greater longitudinal data sample that captures more match-play contextual data. Data collection and availability are still potentially prohibitive of to what extent this can be sought in the short term, but efforts should be made by the practitioners, AFLW clubs, and the league to ensure this issue is alleviated in the near future. Upon this data becoming more readily available, more intensive methodologies, including machine learning, will be appropriate and able to give further insight.

Combining physical, technical, and tactical match-play performance measures in future research may provide a more holistic representation of performance, in line with the 'dynamic systems' approach of performance analysis. Utilising methods that can incorporate all of this information, like neural networks used by *Watson et al. (2020)*, can allow for this representation to become closer to reflecting intertwined aspects of match-play performance demands.

## Critical appraisal

While all but one of the included studies were considered to have a low risk of bias (*Thornton et al., 2020b*), it is important to note the common methodological issue apparent concerning data representativeness. Consistent issues regarding sample sizes and representativeness of results were evident, primarily due to the short time frame of data available. Only three included studies had inter-seasonal data, and five included studies investigated the data of multiple teams, only two of which were on an elite population (*Cust et al., 2019*; *Dwyer, Di Domenico & Young, 2022*). Any analysis of an elite population in women's AF is naturally constricted by only a few relatively short seasons worth of data due to the recent establishment of the AFLW competition, as well as the COVID-19 pandemic causing an unfinished season in 2020 (*Black et al., 2019b*; *Kleyn, 2021*). Exacerbating this issue, the likely evolution of match-play over time further contributes to potential problems in research generalisability. This is in addition to the common constraint associated with team sports, where a population of only one team is often used in research because the directly competitive environment is unconducive to collaboration (*McGarry, 2009*).

This issue of data collection from one team is more pertinent in physical performance research. Tactical and technical variables investigated have the advantage within the

AFL and AFLW of a competition-wide match-play collection service in Champion Data, compared to measures of physical performance that are collected by teams on an internal basis, as evident by the population size of the studies of *Cust et al. (2019)* and *Black et al. (2019a)* which incorporate all matches played across multiple seasons.

While much of this review does call for increased data utilisation and collection, caution must be made on the extent of further data collection. A recent report has highlighted issues concerning the ethics and current practice of data collection for performance analysis in sport, with a noted topic being the sensitivity of data collected on players, as well as a suggestion that data that is collected should be only what is needed as dictated by prior evidence (*Australian Academy of Science, 2022*). The question remains in women's AF, and much of women's sport in general, as to what data needs to be collected; it is essential to note that while data requirements can change over time, simply stopping the collection of specific data might hinder future research without a historical time-series of data.

The present scoping review incorporated a systematic search and screening approach to identify available literature to describe methods of match-play performance analysis employed in women's AF. The critical appraisal of methodological quality performed on included studies strengthened synthesised conclusions regarding match-play performance methodology. Table 6 provides a summary of the key issues and recommendations presented in the discussion of this review. The recommendations provided can assist researchers, sports science practitioners, and data analysts to work towards enhancing performance analysis methods to better understand the physical, technical, and tactical match-play performance in women's AF

## CONCLUSIONS

The key findings of this scoping review display that while the groundwork has begun in physical, technical, and tactical performance analysis in women's AF, there is greater room for exploration when considering all physical performance categorisations of *Johnston et al. (2018)* as well as technical and tactical analysis based on the framework of *Lord et al. (2020)*. Data representativeness was a key issue observed in the critical appraisal, indicating that more longitudinal data collection is required for more robust results. Room exists for the expansion of methods away from traditional statistical approaches, particularly through big data analytics. Junior and recreational levels of play also are yet to be researched. Overall, these findings provide a greater context and guidance to researchers and staff within women's AF on the themes, modelling techniques, and populations that have the potential for future research while identifying the current limitations and issues surrounding appropriate methodological selection to analyse these areas.

## ACKNOWLEDGEMENTS

The authors would like to thank Aine Bagnall for her contribution to the screening and critical appraisal of studies.

**Table 6  Table of key issues and recommendations.**

| Issue | Recommendation |
|---|---|
| **Physical Performance** | |
| **Data** | |
| Data limited by one team collection. | Increased data sharing through a centralised, independent body for research. |
| Thresholds for running speeds differ between analyses. | Development and adherence to standardised running speed thresholds. |
| Confounding factors need greater capture and incorporation. | Inferential statistics used currently only compare physical variables in a univariate manner rather than allowing for these factors to be accounted for. |
| **Methods** | |
| All research has been conducted through inferential statistics which are limiting due to univariate tests in a sport are very dependent on a range of contextual information. | Opportunity for increased use of predictive and non-linear analysis with the collection of additional variables that can better represent the complex, multivariate relationships of match-play performance. |
| **Themes** | |
| Many categorisations of physical performance are not covered. | A broader examination of physical performance categorisations, particularly linking training and physical qualities measured off-field to match-play performance. |
| No junior level-of-play analysis. | A greater understanding of junior players' physical performance will provide more information on player development pathways and inform training practices for junior players. |
| **Technical/Tactical Performance** | |
| **Data** | |
| Current data availability means sample sizes used only create results reflecting a small snapshot in time (*e.g.*, one season). | Greater longitudinal data with reproducible methods implemented to allow constant re-evaluation of key variables as well as allowing for fair comparison over time. |
| Data collection in AFLW not as available/accessible as in men's AFL. | Bring collection protocols and availability in line with the AFL. |
| **Methods** | |
| Inferential stats and basic predictive models applied which can be subject to confounding factors not being accounted for in resulting relationships. | Ability for more intensive machine learning and network analysis models to be applied to better capture the complex interrelationships between match-play performance data. Caveat of greater data collection needed to enable these models and themes of analysis. |
| Some papers predict outcomes using data that can be seen as a derivative of the objective variables (*i.e.*, ratio involving goals) or not using holdout test data as a means of evaluation. | More care in modelling with holdout test dataset used for fair comparison of models and results. |
| **Themes** | |
| Only game actions analysis undertaken currently. | All tactical themes of analysis to be investigated when data allows - both through increased data collected to meet specific theme purposes via more variables, granularity of data, and greater longitudinal data. Game actions also needs re-evaluation over a longer period. |
| Physical, technical, and tactical performance all influence one another. | Use datasets incorporating each of these themes within match-play performance analyses; with the potential to use neural networks to produce the best representation of match-play performance with available data. |

**Notes.**

Abbreviations: AFL, Australian Football League; AFLW, Australian Football League Women's.

### Funding

This work was supported by the Australian Government Research Training Program Scholarship. The funders had no role in study design, data collection and analysis, decision to publish, or preparation of the manuscript.

### Competing Interests

Justin Keogh is an Academic Editor for PeerJ.

### Author Contributions

- Braedan R. van der Vegt conceived and designed the experiments, performed the experiments, analyzed the data, prepared figures and/or tables, authored or reviewed drafts of the article, and approved the final draft.
- Adrian Gepp conceived and designed the experiments, authored or reviewed drafts of the article, and approved the final draft.
- Justin W.L. Keogh conceived and designed the experiments, analyzed the data, authored or reviewed drafts of the article, and approved the final draft.
- Jessica B. Farley conceived and designed the experiments, analyzed the data, authored or reviewed drafts of the article, and approved the final draft.

### Data Availability

All raw data is in the tables as this is a systematic scoping review.

### Supplemental Information

Supplemental information for this article can be found online at http://dx.doi.org/10.7717/peerj.14946#supplemental-information.

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
