# Peer review of "Methods of performance analysis in women’s Australian football: a scoping review"

_PeerJ, doi:10.7717/peerj.14946_

## Round 0.1 · original submission · Major Revisions

Some major revisions are needed for the article.

Best regards,

Reviewer 1 ·

Basic reporting

ABSTRACT
Line 46: the authors could increase the clarity of the objectives. Perhaps they want to detail the methods used for conducting match analysis, characterize the time of studies, or eventually provide future directions. At the moment, it is impossible to understand the specific objectives firmly. The authors could add these particular objectives.

Lines 50-52: The background presents a hybrid approach using rationales for the study objectives and a disclosure of the option for the strategy in this review. I suggest lines 50-52 be presented in the methods section (of the abstract).

Lines 55-56: the authors can list the databases alphabetically. Moreover, any reason for the repeated searches?

Lines 56-58: The PCC or PECOS approach could help to understand the eligibility criteria quickly. Example: which type of athlete? Adults and youth? Exposed to any match-play (friendly or official? And simulated? And small-sided games?)?

Line 58: please add the instrument/tool used for the critical appraisal.

Line 61: I suggest adding the information about the total number of studies searched and then the selected ones.

Line 69: the conclusion starts with the period. However, this is not an item previously described in the objectives and methods of results. The results may present that.
Lines 69-76: although a methodological-basis review, it would be important that the conclusions bring some information about which type of methods have been used and how to improve it. The results are focused on population and practices, while the determination concentrates on other targets.

INTRODUCTION
Lines 96-97: only using machine learning? A description of the different methods would be essential.

Lines 98: how can potential influence coaching? Describe that after the sentence.

Lines 113-115: the authors can provide more details about this framework.

Line 117: since male research holds more research, what about describing them aiming to provide a basis for the current research? This can also justify the strategies used by the authors to categorize the variables later on.

Lines 160-165: the authors could use this more specific objective for updating the abstract.

DISCUSSION

This is only a suggestion, but the context can be different. At the discussion, it would be essential to structure the topics to discuss within the main areas. For example, within the match, technical performance talks about: the context and replicability of the studies; instruments, validity, and reliability; primary outcomes; methods of inference; gaps, and future research. The idea is to use this sub-structure for both physical and technical/tactical.

At the bottom of the discussion, it would be interesting to read a table highlighting the main critical points observed in the review and a second column with recommendations for changing the following original articles. This can enhance the key-home message for scientific and practical communities.

Experimental design

Line 175: please describe why the searches were repeated.

Line 201: does snowballing citation tracking was performed? Moreover, did the authors recruit experts to consult the included articles? Finally, were the errata and article retractions analyzed for any articles included in the review?

Lines 213-214: present the validity and reliability of the instrument.

Validity of the findings

Line 242: It is recommended that a brief in-text description of the flowchart. Is it indicated in the current flowchart? Did the authors perform manual searches in addition to manual ones?

Tables 3 and 4. It is recommended to add information about the instruments used for collecting the data and their validity/reliability. Additionally, information about the study's purpose would help to contextualize the data collection. Eventually, it would also be essential to add information about the period in which data was collected.

Table 3: the running speed thresholds are individualized or standardized? Eventually, this information can be added.

Table 4: Observing more details about which performance variables are being followed is recommended. And the instruments used as well.

I suggest including an evidence-gap map that may provide information about how methods have been used for the aims and focus of the research. This can appear at the bottom of the results or before the critical appraisal.

·

Basic reporting

The review is broad and cross-disciplinary interest and is within the scope of the journal.

The field has been reviewed recently, but this review offers a critical appraisal of eligible studies’ methodologies which differentiates this review from the others: a methods-focused review was conducted instead of a results-focused review.

The introduction adequately introduces the subject and clarifies who the audience is and what their motivation is.

Experimental design

The survey methodology is consistent with a comprehensive and unbiased coverage of the subject.

The sources are adequately cited, quoted, and paraphrased as appropriate.

The review is organized logically into coherent paragraphs/subsections.

There are well-developed and supported arguments that meet the goals set out in the Introduction.

Validity of the findings

There are well-developed and supported arguments that meet the goals set out in the Introduction.

The conclusion identifies well-unresolved questions/gaps/future directions for setting priorities in creating new data for both novel and appropriate research.

Additional comments

This review is valuable evidence that identified gaps in the research that came in the early stage of the game's development (women's and girls' Australian football) and will undoubtedly contribute to the evidence-based development of this young sport.

---

## Round 0.2 · Minor Revisions

From line 182 when accessing the link, an error appears on the names of the authors, ie., Jackson Miller appears on the website, but this name is not in the article. Also, the title of the paper is not exactly as it appears on the website.

With kind regards,
Georgian Badicu
Academic Editor
PeerJ Life & Environment

Reviewer 1 ·

Basic reporting

The introduction was improved. It is now easy to read, and there is a good rationale.

Experimental design

The methods of the articles were improved during the revision stage and are now clear. It is logic and coherent and details the methodological options.

Validity of the findings

The findings are interesting and provide an evidence gap map that can be interesting for the scientific community.

---

## Round 0.3 · accepted · Accept

The paper it can be accepted for publication.


With kind regards,
Georgian Badicu
Academic Editor
PeerJ Life & Environment